# Variability of Net Primary Productivity and Associated Biophysical Drivers in Bahía de La Paz (Mexico)

Rafael Cervantes-Duarte [1], Eduardo González-Rodríguez [2,*], René Funes-Rodríguez [1], Alejandro Ramos-Rodríguez [3], María Yesenia Torres-Hernández [1] and Fernando Aguirre-Bahena [1,†]

1 Centro Interdisciplinario de Ciencias, Instituto Politécnico Nacional, Av. Instituto Politécnico Nacional s/n, Col. Playa Palo de Santa Rita, Apdo. Postal 592, La Paz C.P. 23096, Baja California Sur, Mexico; rcervan@ipn.mx (R.C.-D.); rfunes@ipn.mx (R.F.-R.); mtorresh1800@alumno.ipn.mx (M.Y.T.-H.)

2 Centro de Investigación Científica y de Educación Superior de Ensenada, Unidad La Paz Miraflores No. 334 e/Mulegé y La Paz. Fracc. Bellavista, La Paz C.P. 23050, Baja California Sur, Mexico

3 Departamento de Geología Marina, Universidad Autónoma de Baja California Sur, Carretera al Sur KM 5.5., Apartado Postal 19-B, La Paz C.P. 23080, Baja California Sur, Mexico; ja.ramos@uabcs.mx

* Correspondence: egonzale@cicese.mx

† Deceased.

**Abstract:** The use of information of net primary productivity (NPP) from remote ocean color sensors is increasingly common in marine sciences. The resulting information has been used to explain variations in productivity at different spatio-temporal scales and in the presence of climate phenomena, such as the El Niño Southern Oscillation, and global warming. Satellite remote sensing data were analyzed in Bahía de La Paz (BLP), Mexico, to determine the spatio-temporal variation in NPP. In addition, in situ hydrographic data were obtained to characterize the water properties in the bay. The satellite data agree with in situ measurements, validating the satellite observations over this region. The NPP generally presented seasonal variation with maximum values in winter-spring and minimum values in summer–autumn. The variance explained by NPP from the measured variables was ranked as Chl-a < DEN < SST < PAR < WSC. The highest NPP values generally occurred when subtropical subsurface (SsStW) water was relatively shallow. Due to divergence and mixing processes, this water provided nutrients to the euphotic zone, and consequently an increase in NPP and changes in plankton biomass were observed. The annual trends of the variation in hydrographic data with respect to that of remote sensing data were similar; however, it is necessary to increase the number of data validation studies. The remote sensing and in situ measurements allowed for the main biophysical variables that modulate NPP in different time scales to be identified. The satellite-derived NPP data classifies the BLP as a high productivity zone with 432 g C m$^{-2}$ year$^{-1}$. The use of satellite NPP data is satisfactory and should be incorporated into marine primary productivity studies.

**Keywords:** chlorophyll; sea surface temperature; photosynthetic active radiation; wind stress curl; euphotic zone; Gulf of California



## 1. Introduction

Primary production resulting from the photosynthetic processes of phytoplankton and macrophytes constitutes the main source of organic compounds in the oceans [1], and the global production of phytoplankton has been estimated to be between 52–55 Pg C year $^{-1}$ [2]. The environmental parameters that either stimulate or limit phytoplankton growth are light (i.e., intensity and spectral composition), temperature, salinity, micronutrients, trace lements, and some organic compounds. Indirect factors that affect primary production in the ocean include physical processes that result in water-column mixings, such as winds, tides, and advection or retention due to currents or vortices, in addition to biological factors like zooplankton grazing [3,4]. The methods for evaluating net primary productivity (NPP) are diverse and range from the classic $^{14}CO_2$ fixation method introduced by Steemann-Nielsen [5] to the use of remote sensing satellites [6,7].

Continuous satellite chlorophyll a (Chl-a) data with high spatial and temporal resolution and global coverage have now been available for more than 20 years (since September 1997). In addition, Coastal Zone Color Scanner (CZCS) and Ocean Color and Temperature Scanner (OCTS) data have been available from the 1970s and early 1980s. As a result, studies that use Chl-a data to evaluate processes over different time scales have been published for more than two decades. The quantity of available data has also made it possible to study variations in Chl-a over different spatio-temporal scales, such as over days or kilometers. The high spectral resolution of the ocean color images and availability of the derived products have allowed for satisfactory comparisons with information generated in situ, and comparative studies in coastal water bodies (case 2), including bays and lagoons, have been carried out. However, satellite chlorophyll a (Chl-a) data only correspond to the first optical depth [6].

Behrenfeld and Falkowski [8] proposed the Vertically Generalized Production Model (VGPM) to estimate NPP integrated over the euphotic zone using data from remote sensors. This VGPM is based on phytoplankton responses to environmental conditions in the water column [7]. Over a section of the California Current, VGPM results have been shown to be significantly correlated with in situ data [9,10]. The VGPM is considered to be a good estimate of NPP at both global and regional scales and although it was proposed more than 20 years ago, there are few studies that use the NPP to describe productivity. Most studies use satellite chlorophyll data as indicators or proxies of phytoplankton productivity or biomass [11–13]. In Mexico, to the best of our knowledge, only one study has used the VGPM model for the Gulf of Mexico [14], although other studies have used a combination of in situ measurements and models [15].

Bahía de La Paz (BLP) is located on the southwestern coast of the GC. Due to the low cloud cover that is generally present over the BLP and the warm and dry climate of the region, this bay is an ideal site for utilizing passive remote sensing techniques. Despite the ecological importance of BLP as a foraging, reproduction, and refuge area for pelagic organisms, studies on NPP variability within the bay are scarce. Nonetheless, changes in NPP have been reflected in the survival and recruitment of species of commercial importance for the BLP artisanal fisheries and sport fishing fleet [16]). In addition, the feeding behaviors of whale sharks over dense patches of zooplankton reflect patterns in NPP [17], as does the variability in the abundance of California sea lions. Remote sensors provide an opportunity to continuously monitor oceanographic variables, NPP, and their variability in coastal systems. Information regarding the primary factors controlling the magnitude and temporal variation of NPP will allow for an improved understanding of the life cycles of primary and secondary producers. This, in turn, will facilitate the development of ecosystem management strategies. Thus, the objective of this study was to establish that NPP data for BLP obtained from remote sensors may be used to evaluate the temporal variation in productivity while allowing for a determination of the main biophysical factors that control this variation.

## 2. Materials and Methods

### 2.1. Study Area

Bahía de La Paz presents a variety of oceanographic processes that affect NPP, such as vortices and hydraulic jumps, which operate on different spatio-temporal scales [18,19] and affect organisms from plankton to marine mammals [20,21]. According to Verdugo-Díaz [22], the average NPP in BLP is relatively high (350 g C m$^{-2}$ year$^{-1}$). In addition, meteorological events (e.g., tropical cyclones, and climatic phenomena) in BLP that affect ecological cycles [23,24] also affect NPP. These include El Niño events [25] and the recently identified large mass of relatively warm water in the Pacific Ocean off the coast of North America, known as "The Blob" [19] also affect NPP.

Bahía de La Paz is influenced by the oceanographic conditions of the southern GC and has three points of entry. The main entrance to BLP, Boca Grande, is the largest and deepest and is located in the northern region of the bay. The San José Channel, which is shallow

and narrow, is located to the northwest. The San Lorenzo Channel is the shallowest and narrowest of the three and is located in the southern region of the bay, between Espíritu Santo Island and Playa El Tecolote (Figure 1).

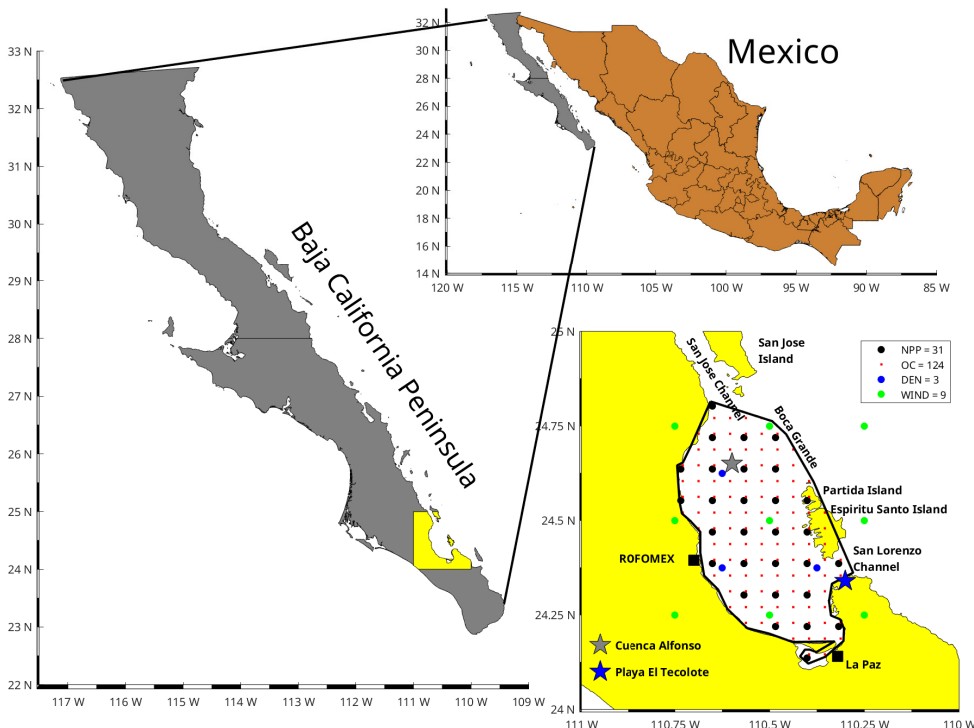

**Figure 1.** Overview of the Baja California peninsula and locations of the satellite data over Bahía de La Paz (BLP). The black circles indicate the pixels with net primary productivity (NPP) data. The red points indicate pixels with data of other variables obtained from Oceancolor (OC). The blue circles indicate density (DEN) data. The green circles indicate wind data. The black squares indicate the city of La Paz and the phosphorous mining complex (ROFOMEX). The black contour line represents the polygon of the study area.

The wind pattern in the bay follows the monsoon pattern that is present throughout the GC. This consists of intense northeasterly winds that begin in June–July and continue through September. These winds change their direction at the end of spring and becoming less intense southeasterly winds that persist through the end of summer. On the other hand, the diurnal variation in local winds is due to the sea-land breeze system, which is correlated with coastal surface currents in the bay. The land breeze reduces the specific humidity over the bay after sunset, which remains low during the night until the following morning [26]. The winds known locally as Coromuel [27] come from the south-southeast during the summer and beginning of autumn with speeds of ∼4 m s$^{-1}$ [26,28].

The water masses present in BLP are Tropical Surface Water (TSW) or Equatorial Surface Water (ESW), Gulf of California Water (GCW), and Subtropical Subsurface Water (SsStW) [29–32]. Sea surface temperature (SST) presents an annual cycle with maximum and minimum values observed at the beginning of September and at the end of February, respectively. At the end of spring, sudden drops in temperature of 1–3 °C may occur that persist for 1–2 weeks, generating superficial pulses of Chl-a. This drop in SST was related to mixing processes produced by the transition from southeasterly to northeasterly winds. During spring and summer, the presence of a marked thermocline and a shallow mixed layer is notable, whereas the depth of the mixed layer is deeper during autumn and winter [29].

The hydrodynamics of BLP are influenced by the cyclonic circulation of the deep zone [30,31,33–35], which results in an elevation of the pycnocline and an increase in nutrient concentrations. These structures produce lateral agitation and mixing in the water

column, which results in vertical isopycnal displacement and the transport of nutrients to the euphotic zone, in addition to concentrating cold water near the surface and causing an increase in sea level [36,37]. Monreal-Gómez et al. [31] considered the vortices to be fertilization mechanisms of primary production.

### 2.2. In Situ Data Collection

The in situ data were obtained from two principal sources. The first source consisted of four oceanographic cruises carried out in January, April, July, and November 2002 onboard the oceanographic vessels Francisco de Ulloa and BIP-XI. The information from these cruises was used to characterize the water masses in BLP with temperature, salinity, nitrate, phosphate, and Chl-a data collected from the surface to 300 m depth [38]. The second source included published papers and postgraduate theses for the period of 1999–2014 that employed temperature, salinity, nitrate, phosphate, and Chl-a data collected in BLP [22,25,29,31,39–42]. A monthly matrix with box-and-whisker plots was generated. Monthly climatologies were generated to describe the intra-annual variability of the aforementioned variables and to compare them at the same temporal scale as that of the satellite data.

### 2.3. Satellite Data

Monthly NPP data, with a spatial resolution of 9 × 9 km, were obtained from 211 (level 3) files corresponding to the last 19 years (July 2002 to January 2020) from the Oregon State Ocean Productivity homepage [43] spanning the Aqua-MODIS available data; the NPP estimation employed the VGPM model [8]. Monthly data for other variables (level 3), including SST, Chl-a, and photosynthetically available radiation (PAR), for the same period with a spatial resolution of 4 × 4 km from the NASA Ocean Color homepage were included [44]. Additionally, monthly density (DEN) data (level 4) from the Copernicus reanalysis product (satellite-based) called MULTIOBS_GLO_PHY_S_SURFACE_MYNRT_0 15_013 [45] with a spatial resolution of 25 × 25 km were obtained for the period of 2002–2018 (204 months) [46]. The monthly zonal and meridional wind components (u and v, respectively) for the period of 2000–2019 (240 months) were calculated from daily data obtained from the Copernicus Climate Reanalysis ERA 5 product produced by the European Centre for Medium-Range Weather Forecasts (ECMWF) [47]. The effect of the wind in the region was evaluated by calculating the wind stress curl (WSC) [48], which is a measure of vertical pumping and is proportional to the vertical velocity. The sign of the WSC refers to its direction, with positive and negative values indicating upwelling (divergence) and downwelling (convergence), respectively [49]. The WSC is defined in Equation (1):

$$WSC = \frac{\partial}{\partial x}\left(\frac{Y_s}{f}\right) - \frac{\partial}{\partial y}\left(\frac{X_s}{f}\right) \cdot m \cdot s^{-1} \cdot 10^6, \tag{1}$$

where $X_s$ and $Y_s$ are the wind components; and $f$ is the Coriolis parameter. This parameter is defined in Equation (2):

$$f = 2 \cdot \Omega \cdot sen(lat) \tag{2}$$

where $\Omega = 7.292 \times 10^5 \cdot s^{-1}$ is the angular velocity of the earth.

### 2.4. Monthly Climatologies from Satellite Data

Given that level 3 variables are global, a polygon around BLP was constructed (Figure 1), and only the pixels within this polygon were included in the study. The number of valid data points for each variable is as follows: NPP (12), SST (24), Chl-a (24), PAR (24), DEN (3), and WSC (3). However, for wind data (i.e., WSC), which require at least 2 data in longitudinal and latitudinal directions, nine data points around BLP were selected (green dots; Figure 1). Monthly and seasonal variations in NPP, SST, Chl-a, PAR, DEN, and WSC were visualized with box-and-whisker plots.

*2.5. Correlations among Satellite Variables*

Time series were constructed from the monthly values of the data within the polygon for all variables, and in the case of WSC, the nearby data (green dots, Figure 1). The relationships between the NPP time series and those of the other variables (i.e., Chl-a, SST, PAR, DEN, and WSC) were evaluated via Spearman correlations (211 monthly dates) at a 95% significance level ($p < 0.05$).

*2.6. Climate Indices*

The multivariate El Niño/Southern Oscillation (ENSO) index (MEI) [50] from 1979 to 2019 and the Pacific Decadal Oscillation index (PDO) [51] from 1854 to 2019 were used to identify the relationships between the selected variables and the interannual and interdecadal signals known for the Pacific.

*2.7. Determination of Periodicity*

The most relevant components of each of the time series were determined according to the cyclic descent technique [52], which was implemented using the created `Periods` function in Matlab® [53]. This technique consists of decomposing a series to detect the most important harmonics (i.e., frequency, phase, and amplitude) that are statistically significant. The sum of all the significant harmonics generates a statistically adjusted model for the series. This technique has the advantage of the frequency of each harmonic being expressed in terms of the unit of time of the study series (months, in this case). Moreover, the harmonics are arranged by the importance of their statistical contribution. This technique was applied to all time series (i.e., NPP, SST, Chl-a, PAR, WSC, DEN, MEI, and PDO) to extract the statistically significant periods.

*2.8. Calculation of NPP and SST Anomalies and Determination of the Warm and Cold Events*

From the NPP and SST time series, the monthly climatologies and anomalies were calculated. For this, the climatological value of the month was subtracted from that of each date in the series [54,55]. To detect warm and cold periods, the regime shift detector (RSD) and/or the temporary synchronization of a regime change for both series were employed. The RSD is useful for identifying abrupt changes in the mean or variance within a time series by a student t-test. In addition, the RSD detects changes with a minimum lag and without bias when the observations are either above or below the mean of a longer interval (red route). The RSD was applied with a significance level of 90% ($p < 0.1$) and the following entry parameters: cut length of 120 months (detects a regime change of at least 10 years) and a Huber weighting parameter of 1. The inverse proportionality method with 4 corrections (IP4) was used for the red road estimation using a sub-sample size of 59. Prior to the analysis, a pre-whitening of the data was carried out [56].

## 3. Results

*3.1. Seasonal Variations in In-Situ Chl-a, Nutrients and Hydrographic Data*

The Chl-a values for the whole series ranged from 0.1–6.91 mg m$^{-3}$, with higher mean values observed from November to June (0.72–1.24 mg m$^{-3}$) and comparatively lower values observed from July to October (0.16–0.66 mg m$^{-3}$). In January, a greater dispersion of the values was recorded compared to that of the other months (Figure 2A). Temperatures showed ample variation between 16–32 °C, with minimum mean values observed in winter (22 °C, March) and maximum values in summer (>28 °C, August and September; Figure 2B). Salinity presented a range of values from 34.1 to 36.4, with an average value of 35.5 and increased dispersion in March and November (Figure 2C). Nitrate presented values ranging from not detected (ND) to 10 μM, with concentrations (average and ±1 SD) observed in January (1.50 ± 1.72 μM), February (2.50 ± 2.40 μM), November (1.94 ± 2.15 μM), and December (0.73 ± 0.51 μM) and the minimum values observed between July and October (0.34 ± 0.59 μM; Figure 2D). Phosphate presented a range of values from 0.08 to 2.42 μM, with values (average and ±1 SD) observed in March

and June (0.95 and 0.93 ± 0.44 µM ± 0.37 µM, respectively), and minimum values observed in September (0.43 ± 0.11 µM; Figure 2E).

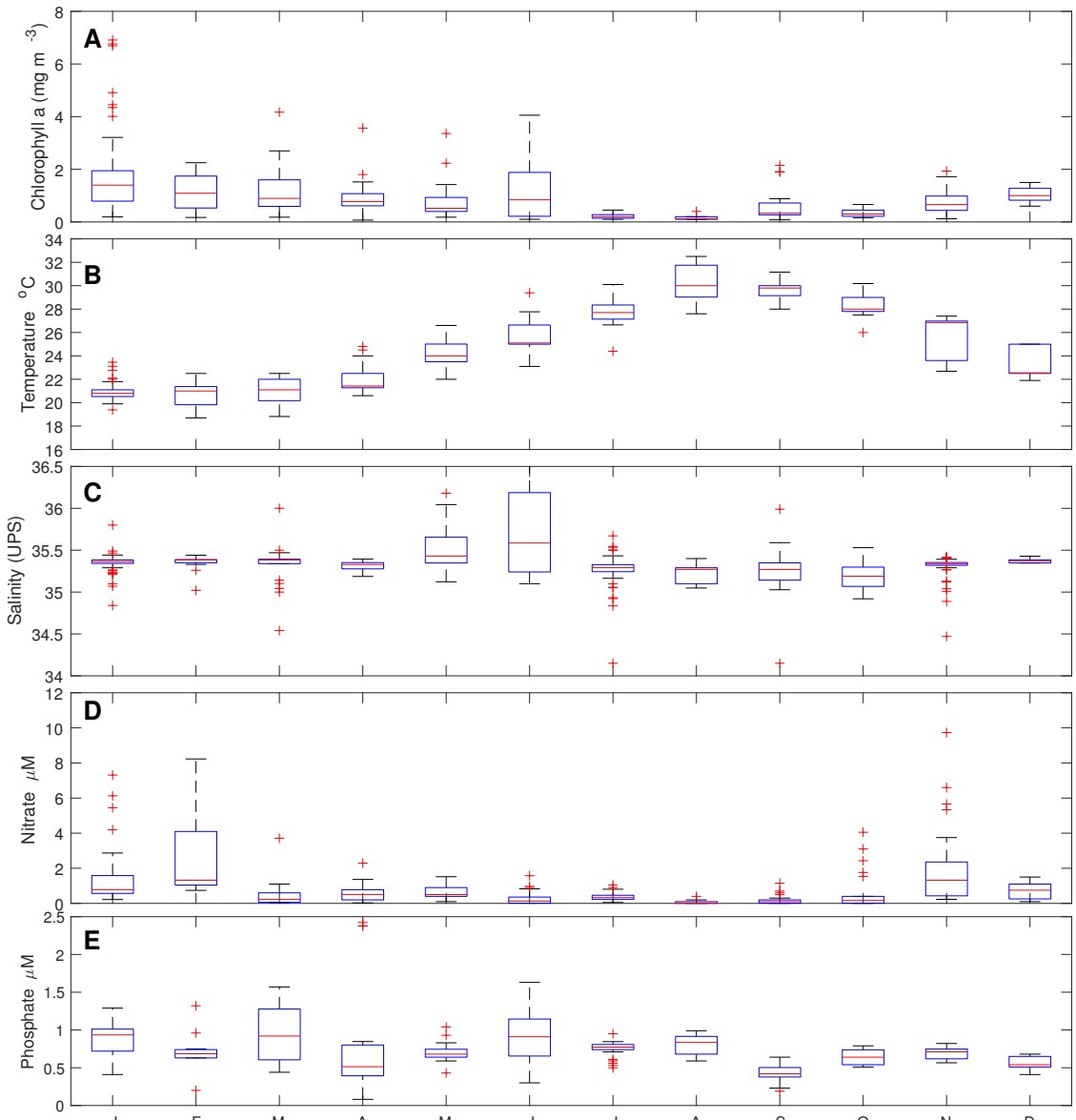

**Figure 2.** Seasonal cycle of in situ surface data (25–75% percentiles, means, and extreme data) recorded in Bahía de La Paz (BLP). (**A**) chlorophyll-a (Chl-a), (**B**) temperature, (**C**) salinity, (**D**) nitrates, and (**E**) phosphates.

The in situ SST data showed that the magnitude and pattern of variation presented an annual cycle, with minimum average values in winter that were related to mixing processes produced by northwesterly winds and maximum values in early autumn and summer associated with increased irradiance. In general, the seasonal trend in Chl-a was greater during the first half of the year, coinciding with the comparatively higher phosphate values observed from winter until mid-summer, although the nitrate concentration increased from mid-autumn to mid-winter.

The average salinity in BLP (35.5) was higher than that of the GC, due to high evaporation (300 mm year$^{-1}$). Additionally, BLP exchanges water masses with the GC, mainly through Boca Grande in the northwestern portion of the bay. The resident water masses in BLP are Gulf of California Water (GCW) in the surface layer and Subsurface Subtropical Water (SsStW) below GCW (Table 1). The limit between both bodies of water can be identified by a decrease in salinity (<34.9) in the water column [29–31]. In turn,

Tropical Surface Water (TSW) flows from the gulf into BLP due to advection by cyclonic vortices and the tidal exchange in the southern part of the gulf. Once there, its salinity increases (>35.0) due to evaporation processes, and the water mass transitions to GCW [31]. SST and salinity follow the same general trend and increase in mid-spring, although salinity decreases in summer and early fall, even when the temperature is maximum during that period (Figure 2).

**Table 1.** Water masses in Bahía de La Paz (BLP) based on data obtained in 2002, GCW and SsStW.

| Variable | Gulf of California Water (GCW) | | | | Subsurface Subtropical Water (SsStW) | | | |
|---|---|---|---|---|---|---|---|---|
| | **JAN** | **APR** | **JUL** | **NOV** | **JAN** | **APR** | **JUL** | **NOV** |
| **Z(m)** | 10 | 10 | 10 | 10 | $106 \pm 17$ | $79 \pm 10$ | $200 \pm 12$ | $178 \pm 12$ |
| **T (°C)** | $20.7 \pm 0.4$ | $20.9 \pm 0.4$ | $25.8 \pm 1.0$ | $27.0 \pm 0.1$ | $15.3 \pm 0.3$ | $14.6 \pm 0.2$ | $13.9 \pm 0.1$ | $14.3 \pm 0.3$ |
| **S** | $35.34 \pm 0.09$ | $35.32 \pm 0.03$ | $35.25 \pm 0.08$ | $35.33 \pm 0.01$ | <34.9 | <34.9 | <34.9 | <34.9 |
| **NO$_3$** | $3.4 \pm 3.0$ | $3.4 \pm 3.1$ | $2.7 \pm 2.5$ | $0.9 \pm 0.7$ | $22.8 \pm 11.7$ | $19.3 \pm 5.3$ | $17.0 \pm 4.2$ | $21.9 \pm 3.2$ |
| **PO$_4$** | $1.0 \pm 0.3$ | $0.8 \pm 0.5$ | $1.1 \pm 0.2$ | $0.7 \pm 0.1$ | $2.2 \pm 0.6$ | $2.3 \pm 0.5$ | $2.5 \pm 0.5$ | $2.7 \pm 0.3$ |
| **CHL** | $1.8 \pm 0.4$ | $1.9 \pm 1.7$ | $1.1 \pm 1.0$ | $0.5 \pm 0.2$ | $0.19 \pm 0.11$ | $0.14 \pm 0.11$ | $0.13 \pm 0.07$ | $0.07 \pm 0.05$ |

*3.2. Seasonal Variation in Satellite-Derived NPP and Biophysical Conditions*

The mean NPP values in May were relatively high in the first semester (January–June) and December (>1000 mg C m$^{-2}$ d$^{-1}$) and decreased from July to November, with minimum values observed from August to October (~500 mg C m$^{-2}$ d$^{-1}$, Figure 3A). From the monthly climatologies, it was determined that 430 g C m$^{-2}$ year$^{-1}$ on average are produced in BLP (see Figure S1 in Supplementary Materials). Chlorophyll a presented a pattern of variation similar to that of NPP in an interval ranging from 0.15–19 mg m$^{-3}$. This was consistent with the values recorded in situ. Higher Chl-a values (>5 mg m$^{-3}$) with the greatest dispersion were observed from January to March, June, and December, while minimum values and lower dispersion were observed in April and from July to November (Figure 3B). The SST values showed an ample range of variation, which was similar to that observed in situ (18–34 °C), with minimum values in winter and early spring (23 °C, January–April) and warm values observed in summer and autumn (28 °C, July–November). A greater dispersion of SST data was observed in June (22–30 °C) and July (24–32 °C; Figure 3C). The PAR data presented an average maximum value in June (63 Einstein m$^{-2}$ d$^{-1}$) and minimum values in January and December (25 Einstein m$^{-2}$ d$^{-1}$), with the greatest data dispersion observed in March, May, and November (Figure 3D). The density of the surface water showed an inverse pattern to that of SST, which reflected the influence of temperature on density (Figure 3E). The DEN values were greatest from autumn to the end of winter, with higher values observed in February and March (1024.5 kg m$^{-3}$) and lower values observed in August and September (1021–1022 kg m$^{-3}$; Figure 3E). The WSC data ranged from −1.0–1.7 m s$^{-1}$, with negative values observed from May to September and maximum positive values observed from January–February and from November–December (Figure 3F).

High NPP values were directly related to high Chl-a and DEN values and positive WSC values, whereas they were inversely related to PAR and SST. Similarly, high NPP values also reflected the availability of nutrients in situ (NO$_3$+PO$_4$) in winter (Figure 2). When the winds in BLP are weak, WSC values are negative (summer and autumn), PAR and SST present maximum values, and DEN begins to decrease. These changes are associated with a tropical water mass, and minimum NPP values are observed between August and November (Figure 3). The monthly climatological maps of the spatio-temporal variation in NPP, CHL, SST, PAR, DEN, and WSC, in addition to the monthly climatologies for each series, can be seen in the Supplementary Material.

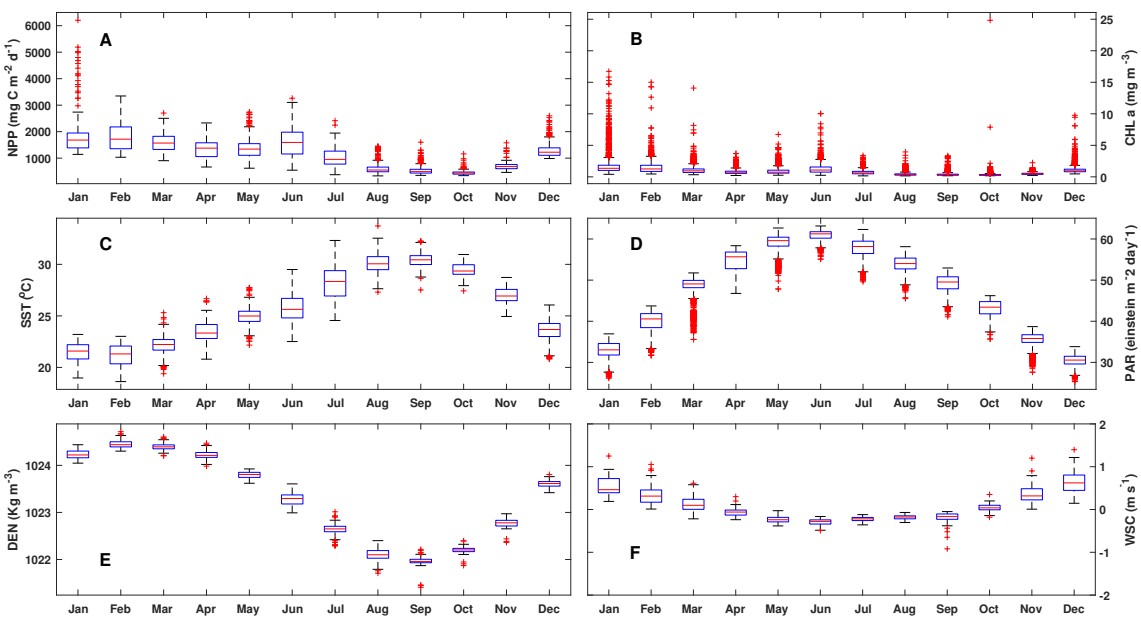

**Figure 3.** Monthly climatologies cycle box plots [25–75% percentiles (box)], 5–95% percentiles (bars), and median and extreme values of: (**A**) net primary productivity (NPP), (**B**) Chlorophyll-a (Chl-a), (**C**) sea surface temperature (SST), (**D**) photosynthetically available radiation (PAR), (**E**) relative surface density (DEN), and (**F**) wind stress curl (WSC).

### 3.3. NPP vs. Environmental Variables

The Spearman correlations between NPP and the environmental variables were significant ($p < 0.05$) and directly proportional to Chl-a (0.96), DEN (0.86), and WSC (0.54) and inversely proportional to SST ($-0.79$) and PAR ($-0.57$). The most important correlations with NPP, as a function of the absolute value of R, were with Chl-a, DEN, SST, PAR, and WSC. Without considering Chl-a, NPP responded to changes in DEN and SST that were associated with the water masses present in the area, PAR (a measure of suspended matter) and wind (WSC).

### 3.4. Interannual Variation and Periodicity of NPP and Biophysical Conditions

The harmonics of the time series indicated three distinct periods for each variable (Table 2). In the NPP series, we identified five significant harmonics with an accumulated coefficient of determination ($R^2$) of 0.66, of which the three most important were 12, 6, and 56 months (Table 2; Figure 4A). The Chl-a series contained four significant harmonics with an accumulated $R^2$ of 0.461, the most important being 12, 6, and 55 months (Table 2; Figure 4B). The SST series showed a marked annual and semi-annual periodicity with six significant harmonics and an accumulated $R^2$ of 0.945, with the most important being 12, 6, and 53 months (Table 2). The PAR series consisted of six significant harmonics and an accumulated $R^2$ of 0.969, with 16, 6, and 75 months being the most important (Table 2; Figure 4D). In the DEN series, seven harmonics were identified with an accumulated $R^2$ of 0.989, with 12, 6 and, 102 months being the most important (Table 2; Figure 4E). In the WSC series, three significant harmonics were identified with an accumulated $R^2$ of 0.878, associated with 12, 6, and 18 months (Table 2; Figure 4F). The MEI series showed 13 significant harmonics, with an accumulated $R^2$ of 0.73, and 66, 133, and 44 months were identified as being the most important (Table 2; Figure 4G). The PDO series (1850–2019), showed a total of 34 significant harmonics and an accumulated $R^2$ of 0.508, with the most important periods being 68, 517, and 34 months (Table 2; Figure 4H).

**Table 2.** Summary of the periods for the first three harmonics for each variable analyzed by the cyclical descent technique. The $R^2$ values is the accumulated correlation coefficient for each variable per period. The numbers in per indicate the total number of significant harmonics for each variable and the final $R^2$ considering these harmonics. Abbreviations: Net primary production (NPP), chlorophyll (CHLO), sea surface temperature (SST), photosynthetically available radiation (PAR), wind stress curl (WSC), relative surface density (DEN), multivariate El Niño/Southern Oscillation Index (MEI), Pacific decadal oscillation (PDO).

| NPP | | CHLO | | SST | | PAR | | WSC | | DEN | | MEI | | PDO | |
|---|---|---|---|---|---|---|---|---|---|---|---|---|---|---|---|
| **Per** | $R^2$ | **Per** | $R^2$ | **Per** | $R^2$ | **Per** | $R^2$ | **Per** | $R^2$ | **Per** | $R^2$ | **Per** | $R^2$ | **Per** | $R^2$ |
| 12 | 0.485 | 12 | 0.245 | 12 | 0.888 | 12 | 0.943 | 12 | 0.823 | 12 | 0.972 | 66 | 0.142 | 68 | 0.066 |
| 6 | 0.59 | 6 | 0.383 | 6 | 0.916 | 6 | 0.963 | 6 | 0.874 | 6 | 0.983 | 133 | 0.274 | 517 | 0.127 |
| 56 | 0.63 | 55 | 0.435 | 53 | 0.922 | 75 | 0.965 | 18 | 0.877 | 102 | 0.985 | 44 | 0.391 | 229 | 0.18 |
| 5 | 0.66 | 4 | 0.461 | 6 | 0.945 | 6 | 0.969 | 3 | 0.878 | 7 | 0.989 | 13 | 0.738 | 34 | 0.508 |

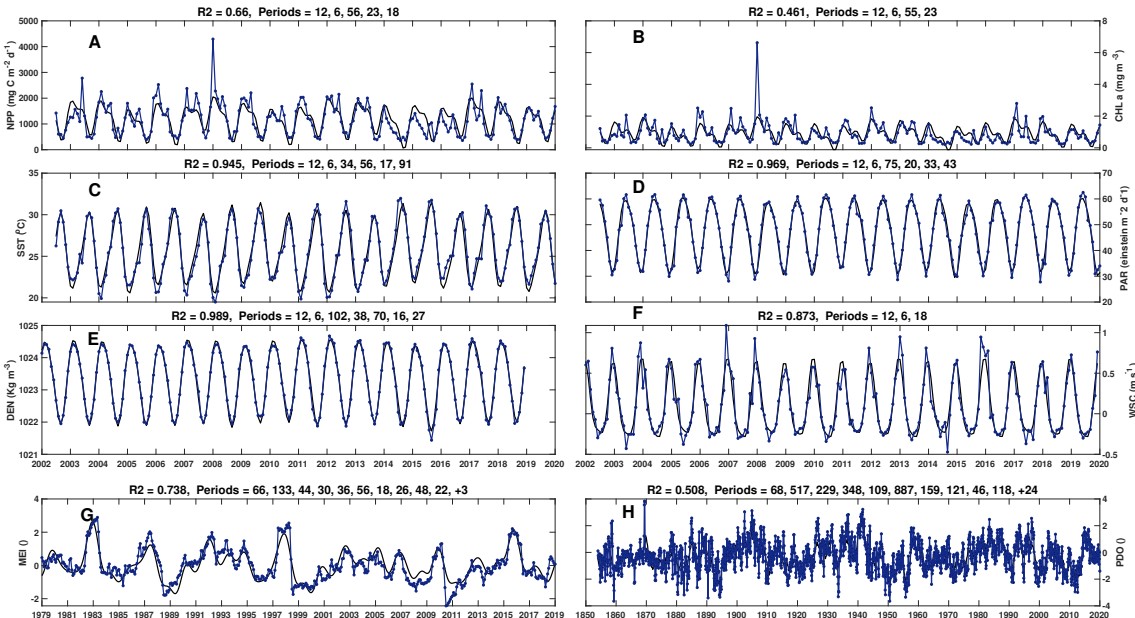

**Figure 4.** Interannual variation of (**A**) net primary productivity (NPP), (**B**) Chlorophyll a (Chl-a) at the surface (**C**) sea surface temperature (SST), (**D**) photosynthetically available radiation (PAR), (**E**) Density (DEN), (**F**) wind stress curl (WSC), (**G**) multivariate El Niño/Southern Oscillation Index (MEI), and (**H**) Pacific Decadal Oscillation (PDO). The blue line represents the data and the black line shows the resulting harmonic model after applying the cyclical descent technique. (**A**–**D**) July 2002 to January 2020, (**E**) January 2002 to December 2018, (**F**) January 2002 to December 2019, (**G**) January 1979 to December 2019, and (**H**) January 1854 to December 2019.

Considering all of the identified periods, the most important were 12 and 6 months, followed by periods between 50–70 months and those larger than 130 months. This indicates that there are environmental phenomena in the BLP that present intra-annual variation, periods of 4–6 years, and periods greater than 11 years.

### 3.5. Regime Shift in NPP and SST

The regime change detector identified two interannual periods with regard to NPP (2002–2020). The first period spanned from July 2002 to September 2013 (133 months, 11.16 years), during which positive anomalies predominated. The second period spanned from October 2013 to January 2020 (6.25 years) with negative anomalies (Figure 5A). The SST anomalies presented intra- and interannual variation, with positive anomalies prevailing as of 2014 and a maximum observed in July 2014 that was three-fold higher with respect to those of other important anomalies (Figure 5B). The regime changes in SST were similar to those detected for NPP, with a regime of negative anomalies between 2002 and

the end of 2014 and positive anomalies that began in February 2014 and lasted until the end of 2020. These positive anomalies were relatively large from February 2014 to May 2016 (Figure 5B). In general, the comparison between SST and MEI anomalies indicated a direct relationship. From 2002–2006, this was an inverse relationship dominated by negative SST anomalies and positive MEI values. The relationship was also an inverse one at the beginning of the maximum anomaly, which occurred in 2014, against negative MEI values (Figures 4G and 5B).

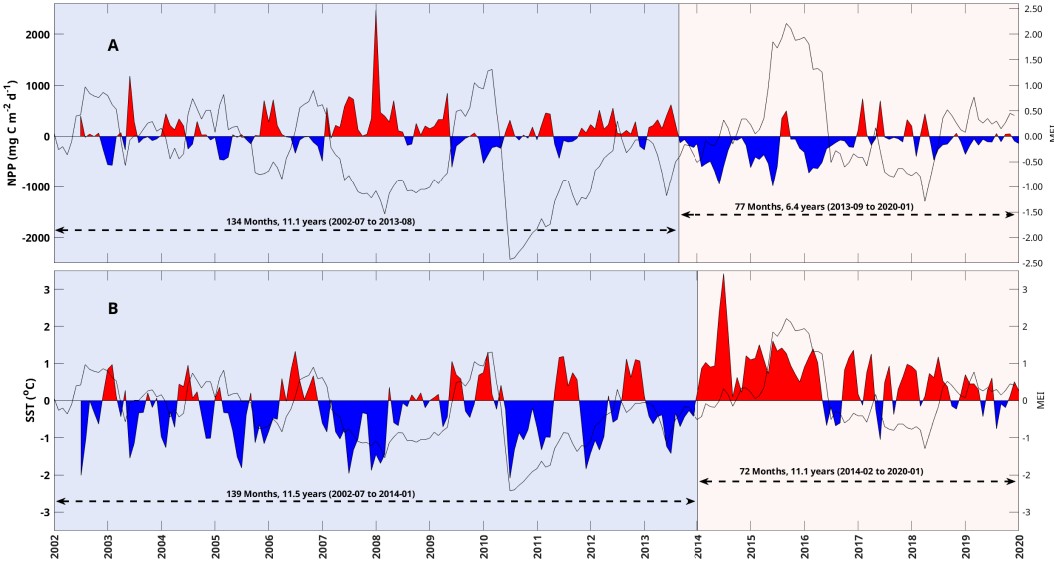

**Figure 5.** Interannual anomalies of (**A**) net primary productivity (NPP) and (**B**) sea surface temperature (SST) during 2002–2020. The gray line corresponds to the multivariate El Niño/Southern Oscillation (ENSO) index (MEI). The blue (cold) period corresponds to the first regime (2002–2007 to 2013–2014) and the pink (warm) period corresponds to the second regime (2014 to 2020).

## 4. Discussion

In general, the intra-annual variation of in situ hydrographic data agreed with that of the satellite data. With respect to the data obtained by remote sensors, the variables (e.g., SST) showed that the magnitude and pattern of variation presented an annual cycle, with minimum average values in winter that were related to mixing processes produced by northwesterly winds and maximum values in early autumn and summer associated with increased irradiance. The average salinity in BLP (35.5) was higher than that of the GC due to the high evaporation in the bay ($\sim$300 mm year$^{-1}$).

In BLP, the hydrological characteristics of GCW throughout the year presented some changes due to local influences (e.g., solar radiation, wind, and tropical cyclones). This was not the case for SsStW, which maintained relatively constant values throughout the year near its borders with regard to its hydrological characteristics. For example, the SST of GCW at 10 m depth was lower in winter-spring compared to that in summer–autumn ($\sim$7 °C difference), while the temperature variation was only $\sim$2 °C at the limit between GCW and SsStW (Table 1). On the other hand, salinity (10 m depth) showed little variation throughout the year, and the values varied around 35.3, which may be characteristic of GCW. Lower salinity values in the water column (<34.9) were directly related to the presence of SsStW.

Some of the processes that bring nutrients to the euphotic zone in BLP are the coastal divergences caused by the circulation around Boca Grande [26], wind mixing [29], and the contributions of anthropogenic wastewater in the southern portion of the Ensenada de La Paz lagoon [57]. The highest concentration of nitrate ($NO_3$) at the surface (Table 1) was related to the winter–spring period, which was in turn related to positive WSC values. Furthermore, the distribution of $NO_3$ in BLP was mainly associated with the high concentrations of $NO_3$

transported by SsStW [58]. Under these conditions, it seems that divergence and mixing in BLP led to high availability of $NO_3$ for primary producers given that in January and April, SsStW was found at a shallow depth (Table 2).

Subtropical Subsurface Water contains a high average concentration of phosphate ($PO_4$; 2.0–3.0 µM) [58], and as with $NO_3$, phosphate can be the main nutrient source in the surface layer during winter-spring. Although phosphate in GCW presented values ranging from 0.15–2.4 µM, its availability throughout the year was maintained ($> 0.5$µM). Thus it does not seem to be a limiting factor for phytoplankton growth. Other sources of $PO_4$ are phosphorite mining in San Juan de la Costa (ROFOMEX) and contributions from the Ensenada de La Paz lagoon [59].

In GCW, the average concentration of Chl-a at the surface increased during winter–spring and decreased during summer–autumn, while the Chl-a concentration was generally low in SsStW ($<0.2$ mg m$^{-3}$), which may be a characteristic of this water mass (Table 1). High subsurface values have been reported in spring and summer due to the accumulation of nutrients at the bottom of the thermocline [22]. The phytoplankton community may be dominated by nanoplankton and not microplankton due to prevailing environmental conditions [42]. The largest phytoplankton fraction is constituted by diatoms in the warm period, followed by the dinoflagellate fraction. The characteristic species of the warm period are *Protoperidinium* sp. and *Nitzchia delicatissima*, whereas the species characteristic of the cold period are *Chaetoceros compressus* and *Coscinodiscus perforatus*, in addition to species common during both periods (e.g., *Thalassionema frauefeldii*) [42].

In general, zooplankton biomass in BLP showed increases ($>300$ mL 100 m$^{-3}$ of filtered water) in winter (January or February) and maximum values ($>500$ mL) in spring [21,60]. Zooplankton biomass ranged from 50 to 1000 mL, and values between 100 and 450 mL were more frequent, with averages between 146–829 mL from winter to spring, with maximum values observed in February, May, and June (556, 829, and 553 mL, respectively) [21,60]. In summer, even when biomass decreased ($<200$ mL), increases of around 300 mL were observed in July [21,60], and similar values inside and outside a cyclonic vortex (500 mL) were observed in Cuenca Alfonso, while decreases in biomass (130 mL) were observed in the southern portion of the bay in June [57]. Increased zooplankton biomass during the warm season and subsequent grazing may have decreased phytoplankton biomass in BLP [61]. However, during the El Niño of 1997–1998, the volume of plankton decreased in summer (50–200 mL), with comparatively higher values ($>150$ mL) occurring near the southern and shallower areas of BLP (Ensenada de La Paz and El Mogote) [62]. During the autumn, although no wide-coverage sampling was carried out in BLP, biomass appeared to be very low ($>50$ mL), despite biomass values of $\sim200$ mL in the San Lorenzo Channel, located to the south of Espíritu Santo Island [63].

The influence of the wind, by means of WSC, increased in the winter and autumn periods and resulted in the vertical pumping of subsurface water, increasing nutrient concentrations at the surface and subsequently NPP [49], which was clearly identified in in-situ data. However, the results suggest that an inverse process occurs in spring and summer in the absence of winds (or with low-intensity winds) and increased solar radiation. These conditions result in the collapse of the thermocline, a lack of nutrients at the surface, and a decrease in NPP in the euphotic zone. Despite this, during the transition of the American Monsoon in June, local breezes and winds known as Coromuel provoke mixing and turbulence that fertilize the water column, at scales of days, which results in a slight increase in NPP. This was also indicated by the remote sensing climatologies that showed a relationship between local NPP, the Coromuel months, and increases in Chl-a in June. However, the temporal scale of Coromuel breezes was not considered in this study, and thus the relationship could not be conclusively confirmed.

The relationship between the increase in NPP and DEN is due to processes that transport dense waters that are generally located near the bottom and rich in nutrients to the surface layer. The inverse relationship between NPP and SST has been widely documented in coastal upwelling areas [11]. The increase in NPP associated with lower SST has also

been related to the origin of the water, as in the case of DEN. When the SST is colder than the surrounding waters it is due to upwelling processes (+WSC) that are modulated by the influence of the wind or changes in bathymetry that facilitate the transport of subsurface water to the surface [31].

The variation of NPP as a function of PAR has been studied based on the photosynthetic efficiency of phytoplankton [22]; however, at high levels of irradiance, the irradiance itself may inhibit photosynthesis. Therefore it is common for NPP to be minimum at the surface increasing towards the base of the euphotic zone during late spring and summer. The analysis of the time series of NPP, Chl-a, SST, PAR, DEN, and WSC showed two important periodicities of 12 and 6 months that corresponded to annual and semi-annual cycles, respectively.

These periods are the result of the seasonal atmospheric dynamics that influence stratification and surface layer mixing in BLP, such as monsoon-type winds that are characterized by presenting dominant north-northwesterly winds during the winter and spring months and south-southeasterly winds in the summer and autumn [26]. The effects of winds on the surface layer of the GC were varied. On the eastern coast of the GC, north–northwesterly winds (5 m$^{-1}$) produce coastal upwelling in winter that transports nutrient-rich subsurface water to the surface. The third period, with frequencies of 55, 53, and 56 months, was recorded for NPP, Chl-a, and SST, and roughly corresponded to the effects of ENSO-type events that decrease NPP and Chl-a when SST increases.

For the MEI and PDO series, the most important periods were 66 and 68 months, respectively, which have been related to the effects of the same event. The RSD showed a cold period of 133 months (11.6 years; blue background, Figure 5) between 2002 and 2013. This period corresponds to the second-most important period of the MEI, which is equivalent to approximately one solar cycle (11 years), and its influence has been described in the records of laminated sediments in BLP [23]. Other frequencies of 75, 18, and 102 months resulted for PAR, WSC, and DEN, although the origin of generation remains unknown.

The time series of NPP and SST anomalies showed annual and inter-annual variation, which has been related to coastal upwelling processes in the GC [57]. In the BLP, these have been associated with local processes related to diurnal winds, breezes, and bathymetric effects [30]. On an interannual scale, SST anomalies have been associated with ENSO climatic events [28]. The interannual overlap of the NPP series with that of the MEI showed high agreement between the positive anomalies and the negative phases of the MEI, particularly between January 2007 and August 2009, in addition to an inverse relationship between negative NPP anomalies and positive MEI values (2002, 2004, 2006, and 2009). Interestingly, anomalous negative NPP values that did not correspond to positive MEI anomalies were present at the end of 2013 and early 2014, along with a significant decrease in NPP and values close to zero of the MEI. This suggests that there are other environmental processes affecting the bay like The Blob.

The Blob was an anomalously warm water mass (~500 km wide and 90 m deep) that developed in the northwest Pacific and spread southward, persisting through 2014 and 2015 [64–67]. The Blob caused anomalous heating in the study area from the end of 2013 until 2016, although it ended in 2015 due to the presence of the ENSO of 2015–2016 [19]. Similarly, the results of the RSD seemed to indicate that the BLP entered a warm phase that lasted for several years, beginning at the end of 2013 with the presence of The Blob that was followed by the ENSO event. The methods used in the present study, such as searching for and analyzing the periodicity in satellite variables, make it easier to relate them to climatic events, such as the ENSO or PDO.

Other authors using satellite-estimated Chl-a data have found similar results to those in this study, which is perhaps because only the University of Oregon [43] currently provides satellite NPP information. For example, Robles-Tamayo [57] used spectral analysis and determined that the annual, semi-annual, seasonal, and inter-annual frequencies affect the variability of the Chl-a concentration in the GC. For BLP, significant relationships between environmental factors and chlorophyll satellite data have been found using

exponential models, with steep slopes and high correlations observed during La Niña events [68].

From the oceanographic information of BLP obtained by remote sensors, it was possible to analyze the variation in NPP and the main biophysical variables that control it. The comparison of sensor and in situ data has yielded similar results over the annual cycle, despite in situ data not being systematically collected over time. For example, Verdugo-Díaz [22] estimated an annual average of 350 g C m$^{-2}$ year$^{-1}$. The estimated value obtained from the satellite data was 432 g C m$^{-2}$ year$^{-1}$, which was 22% higher than that of Verdugo-Díaz [22]. However, we consider that both estimates are similar, taking into account that the in situ data refers to a single point in BLP, while the satellite data correspond to 31 data points within the study area, reflecting the good quality of the satellite data.

The results of the present study show that the variables that most influenced NPP in BLP are chlorophyll, SST, DEN, and WSC. The present study was conducted with data that were collected monthly, and thus NPP variation in higher frequencies associated with lunar, tidal, and wind cycles could not be detected. It is advisable to use data collected at higher frequencies (e.g., weekly or daily) to identify this type of variation. It would also be advisable for agencies, such as the National Oceanic and Atmospheric Administration (NOAA) or European Space Agency (ESA), to provide NPP data with higher spatial and temporal resolution to the public as quickly as possible. At the moment, only the University of Oregon provides data with a spatial resolution of 9 × 9 km and 8-day or monthly frequencies that are published with a delay of several months.

The use of NPP satellite data to describe the productivity of a region is unusual since satellite-estimated chlorophyll data are often used as an indicator of productivity and even in reconstructions or NPP models [69,70]. In the case of Mexico, this type of data is rarely used. Instead, generally in situ measurements [71–73] or a combination of in situ measurements and satellite data are used for chlorophyll or other products, such as other indicators of NPP [13,14]. The contributions of this study reflect the importance and potential of using NPP instead of Chl-a concentrations to better understand the role that carbon plays in ocean productivity.

## 5. Conclusions

This is the first study to analyze primary productivity in BLP using remote sensors. The results obtained in this study are consistent with those of other studies that have used chlorophyll data. Net primary productivity data can be used in conjunction with chlorophyll a data, as has been done for more than 20 years. In both cases, in situ studies must continually generate data for comparison. The methodology used in this study was satisfactory for the analysis of NPP in BLP. It also allowed for the level of productivity to be determined as well as the main variables that modulate it, namely Chl-a, SST, DEN, and WSC. The productivity level classifies BLP as a highly productive region, while the variation in NPP is dominated by semi-annual and annual periodicity that is influenced primarily by the wind, which generates mixing and local upwelling that transport nutrients to the euphotic zone. This vertical pumping of water increases Chl-a and decreases SST. On a larger scale, the variation in NPP is influenced to a greater extent by SST and by phenomena like El Niño events, The Blob, or a positive SST regime that results in positive SST anomalies and negative NPP anomalies. The accumulation of satellite data over time to estimate NPP and time series analysis can help shed light on the relationships between variations in productivity and climate change, especially over large time periods like solar cycles (11 years), and may help to explain the impacts of recent anomalous warming events like The Blob in the northeastern Pacific. It is recommended that the proposed methodology be used in similar regions and for the databases to be continually updated in order to ensure greater confidence in the resulting periods.

**Supplementary Materials:** The following are available online at https://www.mdpi.com/2072-429 2/13/9/1644/s1. Figure S1: Monthly climatology time series; Figure S2: Monthly climatology maps of NPP; Figure S3: Monthly climatology maps of SST; Figure S4: Monthly climatology maps of WSC; Figure S5: Monthly climatology maps of DEN; and Figure S6: Interannual monthly anomalies of NPP, SST, WSC and DEN.

**Author Contributions:** Conceptualization, E.G.-R. and R.C.-D.; methodology, E.G.-R., R.C.-D., F.A.-B., M.Y.T.-H. and A.R.-R.; software, E.G.-R.; validation, E.G.-R., R.C.-D and A.R.-R.; formal analysis, E.G.-R.; investigation, R.C.-D.; data curation, E.G.-R.; writing—original draft preparation, R.C.-D., E.G.-R., F.A.-B., and A.R.-R.; writing—review and editing, E.G.-R., R.C.-D., A.R.-R.; visualization, F.A.-B.; supervision, F.A.-B, R.F.-R.; project administration, E.G.-R.; All authors have read and agreed to the published version of the manuscript.

**Funding:** This research was supported by an internal project from CICESE awarded to E.G.R [project number 691110].

**Institutional Review Board Statement:** Not applicable.

**Informed Consent Statement:** Not applicable.

**Data Availability Statement:** Not applicable.

**Acknowledgments:** To SERVANT laboratory from CICESE ULP for their support.

**Conflicts of Interest:** The authors declare no conflict of interest.

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
