# Peer review of "Variability of Net Primary Productivity and Associated Biophysical Drivers in Bahía de La Paz (Mexico)"

_remotesensing, doi:10.3390/rs13091644_

Round 1

Reviewer 1 Report

The content, data and experiment of this paper have been more detailed, and it is recommended to be accepted.

Reviewer 2 Report

The authors seem to have responded to most of the reviewer feedback. The reviewer had asked to highlight the motivation and limitations of the study along with changes in datasets presentation and reorganisation of results and discussion sections. The authors did that to some extent and made efforts to improve the English. The authors have also addressed the reviewer's specific comments in detail. Few additional minor corrections to take into consideration are listed below:

Lines 13-15: Correct the typos and split the sentence in two.

Line 16: Remove ‘oceanographic information analysis’

Line 66-67: remove the statement starting with “when”

Line 114: remove collection

Line 148: Add ‘(MEI)’ after index

Line 149: Add ‘(PDO)’ after index

Line 387: Chl-a SAT is undefined in the text

Lines 415-416: Last sentence in the discussion. This is the first time that marine carbon biogeochemistry is mentioned. The statement should be toned down or enriched by some further details on the links from NPP to marine carbon biogeochemistry.

Line 428: Remove “that”

Reviewer 3 Report

My first review of this article was generally positive and I am not changing my mind. The authors reacted to my comments on the first version of the article and corrected the indicated errors.

Reviewer 4 Report

The manuscript can be accepted.

Author Response

This manuscript is a resubmission of an earlier submission. The following is a list of the peer review reports and author responses from that submission.

Round 1

Reviewer 1 Report

The paper focuses on the variation in primary productivity in the Bay of La Paz, Mexico, its relantionship with environmental variables. The paper carried out statistical analysis based on the information obtained from remote sensing data and the topic is of great significance. However, there are are several serious problems as follows.

  1. The content of the abstract is redundant, and the expression of the key issues is not prominent enough, which needs to be improved.
  2. The introduction don’t provide sufficient background and lack of up-to-date references (recent three years).
  3. The innovation of the paper needs to be improved. This paper only uses a simple statistical method for comparative analysis, and the highlights of this research are not prominent and the structure is not clear.

Reviewer 2 Report

General comments

 The authors analyse the net primary production (NPP) and its biophysical drivers in the Bay of La Paz using satellite observations in synergy with in-situ data. The study demonstrates also regime shifts in the satellite derived NPP associated with SST anomalies and ENSO-MEI oscillations, which is interesting for the predictability of the productivity in this coastal region. The subject addressed is within the scope of the journal. The paper seems to present novel results, and convincingly conclude on how physical processes (divergence and mixing) drives an increase in NPP. The methods used are sound and the figures are well presented. However, the manuscript requires some modifications before being suitable for publication. Below is summary of my key points and some minor comments:

- The motivation for the study should be better presented in the introduction. For instance, why is it so important to determine the NPP over this particular region? What are the potential socio-economic impacts?

- The datasets are poorly presented (sections 2.2 to 2.4), which requires rearranging and additional information (see specific comments). Also, it is not clear why the study is limited to 2002-2020 when satellite data are available for longer period of time. The shortest being Chl-a, which still cover 1997 to present?

- The abstract is missing a closing statement that explains the direct applicability of the results. The introduction needs to be more focused to clearly bring out the knowledge gaps. Similarly, the conclusion lacks a statement that bring the readers back to a more global context and on the way forward (new hypothesis or suggested research avenues).

- Results section is too descriptive. Although the main changes in timeseries are reported and highlighted, the scientific interpretation is rather poor. I also noted that some results appear in the discussion. I suggest for the authors to restructure these two sections for good flow and clarity. The future directions and research recommendations and the limitations of the study are also missing.

- The text is understandable, but the English has to be improved. Some sentences are badly expressed or too verbose, and their meaning is sometimes unclear. The present tense should be used throughout the text when analysing the results. There are some typos, so I recommend for the authors to carefully read the revised version and avoid mistakes.

Specific comments

Title: The title is not concise. Additionally, in the study in-situ data are used in synergy with satellite observations. So, I recommend removing “the application of remote sensors” as it can be misleading and changing the title to: “Variability of primary productivity and associated biophysical drivers in the Bay of La Paz (Mexico)”.

Line 1: Remove “The possibility of” and “and associated products”

Line 3-4: Change the end of the statement to: “...during climate disturbances such as El Niño Southern Oscillation and global warming.

Line 5-9: Badly expressed! please rephrase.

Line 17: Remove the sentence starting with “The remote sensing....”. It reports on general methods which is not necessarily needed in the abstract.

Line 18:  It is a known fact that the satellite and in-situ observations complement each other. Your analysis revealed here that satellite data agrees with in-situ measurements, which indicate the validation of satellite observations over this region.  So, please rephrase accordingly and to keep a good flow, move this point to Line 9 after “in the bay”.

Line 25: change factors to “parameters”

Line 38: ..., tides, advection and ...

Line 39: Add “and phytoplankton biomass”

Line 31: remove “the current”

Line 32-34: Badly expressed! Rephrase please

Line 36: “a few years ago” is misleading. Satellite Chl-a are available at high spatial and temporal resolution continuously with a global coverage for more than 25 years now (since Sep 1997). This, in addition to the CZCS and OCTS data available in 70s and early 80s.  There are plenty of studies that used Chl-a over different time scale for more than 2 decades now. Rephrase please

Line 37: of ocean colour images

Line 38-39: unclear meaning. Clarify!

Line 40-42: Bad expressions. Rephrase

Line 44: Change “by means of” to using

Line 43-44: Sentence too verbose. Split and make it more concise

Line 48: and over a part of ...

Line 53: Add a reference here

Line 55: in coastal systems have allowed

Line 56: Provide references!

Line 57: change ‘was’ to ‘is’. Bahia de La Paz was already used in the previous paragraphs. So please define the BLP acronym when you first mention Bahia de La Paz  in the text.

Line 58: ... the main biophysical factors

Lines 50-58: The link to the objective at the end of the introduction is weak. Focus on the ecological importance and expand further on the motivation of the study. Why is it so important to determine the NPP over this particular region? What are the potential socio-economic impacts?

Line 61-64: Badly expressed and long sentence. Consider splitting and rephrasing.

Line 65: “as due atmospheric disturbances” Grammatically incorrect and the meaning is unclear. Clarify please!

Line 66: define briefly what the Blob is

Line 69: You use the word portion twice here and elsewhere in the text. You could make use of synonyms to avoid useless repetition.

Figure 1 caption: The caption is not very informative on the figure content. Choose another opening title, something like “Overview of Baja California Peninsula and locations of satellite data used over the BLP.” 

There is no need to add the data providers. So please remove those.

The remaining sentences are too verbose. Rephrase and be more concise.

What does the Blue contour line represent?

Line 73-76:  It would help to specify the months when describing the monsoon winds.

Line 81: are the Topical Surface ....

Line 84: Same here, specify the months

Line 85: was related ...

Sections 2.2-2.4:

It seems that some of the data used is in-situ and others are satellite derived? If so, please create 2 sub-sections to describe separately the in-situ from the satellite data used. Also, give information on the data providers and links to where the data can be downloaded from.

Specify why you choose 2002-2018 as your study period?

Are SSS and DEN obtained from satellite derived information? Give more details

Merge section 2.3 with the new in-situ data section. Specify why you calculate a climatology?

Merge section 2.4 with the new satellite data section. Specify why you calculate a climatology?

Include the WSC info (Lines 108-111) in the satellite section and add the WSC equation used.

Lines 97-100: Split the sentence in two.

Create a new sub-section named climate indices and move there the lines 111-113 about ENSO and PDO.

Section 2.5:

This doesn’t need to be a section. Remove the last sentence (The MEI and ....). Move the WSC info to the new satellite data section.

Move the info about how the rest of the timeseries were obtained to when you first mention these data (i.e., new in-situ section or new satellite data section).

Line 139: Calculation of NPP and SST anomalies and determination and warm and cold events.

Lines 151-153 (section 2.8): Merge this with the results.

Line 155: Present clearly the content of the section. Here seasonal variations of in-situ variables are reported. I suggest using “Seasonal variations of in-situ Chl-a, nutrients and hydrographic data

Line 169: Present clearly the content of the section. Here seasonal variations of satellite variables are reported. I suggest using “Seasonal variations of satellite derived NPP and biophysical conditions”

Line 188: Highlight clearly the content of the section. Here interannual variations and periodicity are reported. So, change the title to: “Interannual variations and periodicity of the NPP and biophysical conditions”

Section 3.3: Expand the analysis / report further information on the year-to-year variations shown in Figure 4

Line 188: Highlight clearly the content of the section. Here you analyse regime shift in NPP. So, I suggest using “regime shift in NPP” as a title.

Line 218 (section 3.5): This separation of the section may not be necessary. I suggest you cover it as part of the remaining results subsections.

Discussion:

Some statements and Table 2 (which is a result) appear in the discussion (section 4), which would be more appropriate in the results section. Specifically, Lines 225-244; 286-301; 315-320; 329-351 and Table 2.

The remaining text in the discussion will need to be reorganised accordingly

What are the likely future directions and research recommendations, especially in light of ENSO or PDO type events?? etc.  What were the limitations of your study?

In the discussion section, it is not needed to separate in 4.1, 4.2. After restructuring this section with Results section, just discuss the results without separation.

Figure 2 caption: Seasonal cycle of in-situ surface data .....

Figure 3 caption: Seasonal cycles box plot ....

Figure 4 caption: interannual variations of ....  and specify the starting and ending years

Conclusions:

You need to sufficiently answer the “so what?” about your findings. You seem to be repeating some of your discussion points.

Line 356: What do you mean by marine communities? Do you mean species? The sentence sounds like on a finding from another study. Please clarify and rephrase.

The conclusion lacks a good closing statement. Tell the reader how your findings can be generalized to other regions. You need to bring the readers back to a more global context. Point us on the way forward as well…any new hypothesis or suggested research avenues etc

Reviewer 3 Report

The aim of the work is to determine the patterns of temporal variability in primary productivity in the Bay of La Paz, Mexico, based on the analysis of observational data collected over the past two decades, as well as to identify the causes of this variability. The authors successfully achieve this aim using remote sensing data, as well as in situ data obtained during oceanographic surveys in the Bay. The datasets of net primary productivity (NPP), sea surface temperature (SST), chlorophyll-a (Chl a), photosynthetically available radiation (PAR), wind stress curl (WSC), sea surface salinity (SSS) and density (DEN), covering the period from 2002 to 2020, are analyzed.

Long-term time series are subjected to statistical analysis, as a result of which the annual and semiannual periodicity of NPP and other important environmental parameters responsible for the NPP variability are revealed. In addition, the authors obtained a number of other important results. For example, a high correlation was found between NPP and the environmental variables. Net primary production directly proportional with Chl a, WSC, and DEN and inversely correlated with the SST and PAR.

Substantiating the conclusions of the work, the authors draw on an extensive list of references, which demonstrates their deep understanding of the topic. The article is written in clear language, well-executed with illustrations. I cannot find any weak points in the presented work and, on the whole, qualify it as an interesting article.

Small remarks:

  1. Figure 1 shows the boundaries of the study area. Why are the NPP pixels (black circles) shown outside the Bay of La Paz? Indeed, the work analyzes the marine data of the NPP.
  2. In the phrase on page 2, line 55 “… .in coastal aquatic systems have alloweds for the establishment…” there is a typo in the underlined word.

Reviewer 4 Report

Please find attached comments.
